# Cost-effectiveness requirements for implementing artificial intelligence technology in the Women's UK Breast Cancer Screening service

Armando Vargas-Palacios [1,2] ✉, Nisha Sharma [3] & Gurdeep S. Sagoo [1,4]

The UK NHS Women's National Breast Screening programme aims to detect breast cancer early. The reference standard approach requires mammograms to be independently double-read by qualified radiology staff. If two readers disagree, arbitration by an independent reader is undertaken. Whilst this process maximises accuracy and minimises recall rates, the procedure is labour-intensive, adding pressure to a system currently facing a workforce crisis. Artificial intelligence technology offers an alternative to human readers. While artificial intelligence has been shown to be non-inferior versus human second readers, the minimum requirements needed (effectiveness, set-up costs, maintenance, etc) for such technology to be cost-effective in the NHS have not been evaluated. We developed a simulation model replicating NHS screening services to evaluate the potential value of the technology. Our results indicate that if non-inferiority is maintained, the use of artificial intelligence technology as a second reader is a viable and potentially cost-effective use of NHS resources.

Breast cancer is the most common cancer occurring in women and the most common cancer overall around the world[1]. The UK is not an exception with over 55,000 cases every year and 11,000 annual deaths, breast cancer is the most common cancer in the UK[2]. However, due to improvements in earlier diagnosis and treatment, 76% of patients now survive 10 or more years[2]. In the UK, the earlier diagnosis is attributed to the 2-week wait cancer diagnosis pathway and the UK National Health Service (NHS) national breast cancer screening programme, which has been previously shown to be cost-effective[3–5].

Currently, the reference standard in the UK is for mammograms to be independently double-read[6]. This process whilst improving quality is a labour-intensive process now under pressure due to a radiology workforce crisis as almost all NHS radiology departments subcontract a proportion of their service to a Locum at a high cost[7].

New artificial intelligence (AI) technology could safely and effectively allow programs to move from a double-reading human reader model to a single-reading human reader and an AI reader model safely and effectively. McKinney 2020, determined that in the UK the AI system maintained non-inferior performance as a second reader[8,9]. Kheiron Medical Technologies has developed a machine-learning system called mammography intelligent assessment or Mia®, to perform the role of a second reader and assist radiologists by identifying potential breast cancers on mammograms[10].

A new AI technology such as Mia® must not only be shown to be cost-effective when compared against standard practice (SP) but ideally be cost-saving (or at least equally effective and less costly than SP) to be adopted within a publicly funded healthcare system such as the UK NHS. AI technologies may directly offer several advantages, such as allowing for a screening service to be less labour intensive. However,

[1]Academic Unit of Health Economics, University of Leeds, Leeds, UK. [2]Centro de Investigación en Ciencias de la Salud, Universidad Anáhuac, Mexico, México. [3]Leeds Teaching Hospital NHS Trust, Leeds, UK. [4]Population Health Sciences Institute, Faculty of Medical Sciences, Newcastle University, Newcastle upon Tyne, UK. ✉e-mail: a.vargas-palacios@leeds.ac.uk

these advantages may not be sufficient to show the latter; especially as such systems usually require an initial investment that may not be considered a priority for some trusts.

In this study, we will assess the requirements in terms of price, set-up, maintenance costs, and the need to outsource privately a proportion of the service, combined with the effectiveness of the Mia® AI system to determine what is required for such a technology to be cost-effective for use in the NHS. We will compare SP (two human double-readers) versus single-human reader plus Mia®, in the diagnosis of breast cancer within the NHS national breast screening programme from the perspective of the NHS.

## Results

A discrete event simulation (DES) model was developed to estimate the cost-effectiveness of the Mia® AI technology. The model was constructed to recreate the breast cancer screening process in the UK, while also capturing the natural history of breast cancer. The model adopts the healthcare and social care perspective, follows a cohort of 100,000 women from age 19 for life, and uses quality-adjusted life years (QALYs) as the health outcome measure. The analysis follows NICE guidelines and determined cost-effectiveness using the NICE suggested threshold value (£20,000 per QALY gained) by which the analysed intervention is cost-effective if its incremental cost-effectiveness ratio or ICER (($\text{Costs}_{\text{Mia}} - \text{Costs}_{\text{SP}}$)/ ($\text{QALYs}_{\text{Mia}} - \text{QALYs}_{\text{SP}}$)) is positive and below £20,000, its net monetary benefit (NMB) (($\text{QALYs}_{\text{Mia}}*£20,000 - \text{Costs}_{\text{Mia}}$)) is higher than those of SP or its incremental NMB (INMB) ($\text{NMB}_{\text{Mia}} - \text{NMB}_{\text{SP}}$) is positive.

The NHS national breast screening programme consists of two independent readers evaluating each mammogram. If both assessments match (readers 1 and 2), the outcome will either be normal or abnormal. If the outcome is normal the woman will be discharged and invited to attend screening in 3 years. If the outcome is abnormal the woman will be invited for second-stage screening which involves triple assessment (clinical examination, further imaging ±biopsy). This is referred to as recall. If the results of both initial assessments do not match (readers 1 and 2 disagree), referred to as discordance, then the mammogram will be sent for arbitration. Here, an independent reader or readers will assess the mammogram and decide if normal or abnormal and the previously mentioned processes take place. All recall cases undergoing second-stage screening will either be cancer diagnosis (true positive) or normal/benign findings (false positive). Fig. 1 shows the basic model structure. The model starts simulating a woman from the age of 19 before her invitation to their first breast screening appointment (which occurs approximately at age 50). The entry point (mammogram in Fig. 1) represents women arriving at the services and having a mammography performed. Two views of each breast (4 images) are taken to be evaluated independently either by the Mia® AI system plus a single reader or by a radiologist/radiographer as a first reader and second reader using the two-reader model. From here the model follows the same pathway as the actual service.

The model allows for cancer genesis to occur before each woman's first scan invitation at age 50, by estimating its development and progressive growth starting at age 19. As such, cancer inception can occur at any age before the screening age (50) or in between scans and may progress in size and stage[3,11]. Size was linked with stage of diagnosis using the Nottingham prognostic index (NPI) and the probability of being non-invasive or advanced[12].

Scan accuracy was measured by the sensitivity and specificity of the independent readers and the Mia® technology. Data were obtained from a retrospective multi-center clinical investigation aimed at evaluating Mia®'s performance in a real-life setting. The study was carried out in seven European breast cancer sites representing four centres (three from the UK and one in Hungary). Data were based on the 1-year sample including over 45,000 cases in 2015[10].

Sensitivity was adjusted to account for breast density but was assumed equal for all tumour sizes. Each woman's breast density was determined randomly using the Volpara Density Group (VDG) 1–4 classification based on Wanders et al.[13,14].

Once the cancer is detected, the model does not individually follow patients through their treatment and progression, but estimates incurred costs and QALYs based on their NPI stage at diagnosis and on the average survivability at detection. Costs per year of treatment were based on estimates by Hall et al.[15] and updated to 2020 GBP[16]. QALYs were estimated using the reported utility values by stage of diagnosis and survivability in Hall et al.[15], Rautalin et al.[17], Gray et al.[3], and Fong et al.[18].

The model considers the costs incurred by the NHS for running the service as SP but also when Mia® substitutes one reader. Costs of the use of Mia®: price per scan, set-up, and maintenance costs were obtained from Kheiron Medical Technologies. The base case analysis was performed assuming a 100,000 women population cohort followed through their lifetime.

The results of the model (as well as the scenario analysis) were subject to first and second-order uncertainty analysis to account for uncertainty raised by the source of data and the model structure and methodology used. This uncertainty is assessed by a Jackknife confidence interval for the ICER, a scatterplot, and a cost-effectiveness

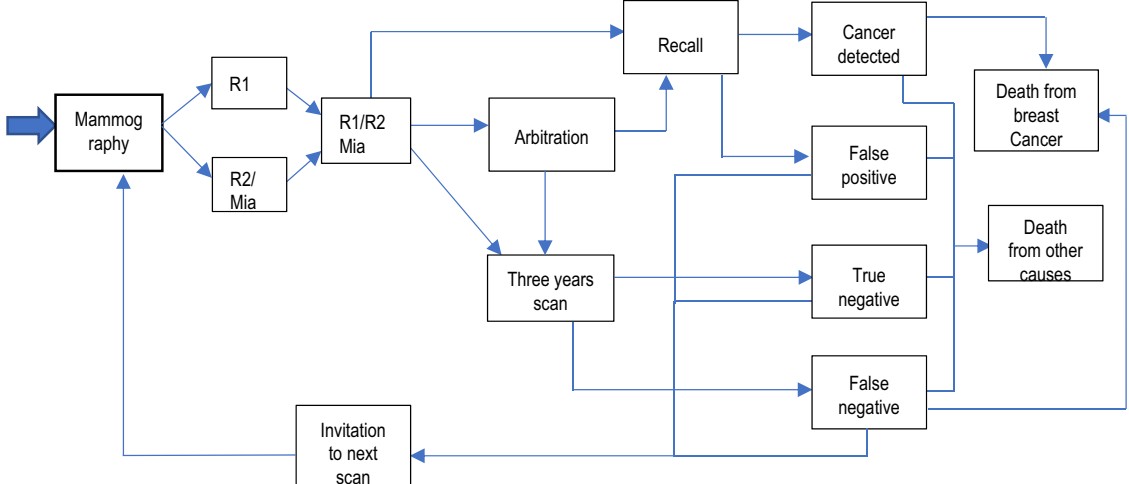

**Fig. 1 | DES model structure.** Schematic of the developed DES model for both interventions. R1 and R2 refer to radiologist readers 1 and 2, respectively. Mia® refers to the AI technology as a second reader. When Mia® is being analysed the second reader is replaced by the AI technology.

acceptability curve (CEAC) to assess the probability of a given intervention being cost-effective at given NICE threshold values.

### Model setup: price, set-up costs, maintenance fees, sensitivity, and specificity
The base case scenario is based on a potential pricing structure in discussion with Kheiron Medical Technologies to estimate cost-effectiveness: one-off set-up cost (£35,000); price-per-scan of £4.72 including maintenance fee. These costs were compared against a calculated cost-per-scan performed by a Radiologist of £5.90 (including overheads and training)[19]. As such, the cost per reading while using Mia® is £10.62; while the cost of two human readers is £11.80. The sensitivity and specificity were: Mia®: 75.1% and 97.3% respectively versus 76.1% and 97% for SP[10].

### Costs and quality of life-estimation
The base case scenario estimates show a negligible difference in costs and QALYs between both interventions. Costs favour Mia® (£4.89 cheaper) but QALY differences favour SP (0.00011 or less than an hour of full health per woman across their lifetime). These estimates produce an ICER above £20,000 and a slightly higher INMB in favour of Mia® suggesting that Mia® is a cost-effective strategy. However, due to the negligible difference in terms of costs and QALYs, these results are highly uncertain as we cannot rule out that these differences are due to chance (Jackknife 95% confidence interval (CI) on the ICER), suggesting that both interventions are likely to be equivalent (Table 1 and Fig. 2).

### Impact of the price per-scan of the Mia® technology
Based on the slight advantage in terms of cost towards Mia®, a maximum reimbursable price (MRP) up to £5.50 would be a viable option as at this price the INMB will be £0 for both interventions, maintaining the likely indifference between the two interventions (Supplementary Data 1).

### Impact of Sensitivity and Specificity
An alternative scenario based on preliminary results of the performance of Mia® where its sensitivity was 88% vs 92% for SP and a specificity of 91% vs 94% against SP (a difference of 4.5% and 3.5%, respectively). Table 2 shows that no price per scan at these levels will make Mia® a cost-effective strategy. Conversely, an increase in the specificity of Mia® of 2% (maintaining specificity of SP at 97% and the sensitivity as the base case for both), allows the MRP to increase to £8.85 (an 87% increase over the base case price (£4.72). If, however, Mia®'s sensitivity is increased by 5%, results remain mostly unchanged (Table 3).

### Set-up costs
In this scenario, we explore the influence of the set-up costs on the average cost per patient. We found this to be minimal as going from an initial cost of £35,000 to £0 only reduces the total cost by less than £0.50 per patient, with minimal impact on the base case scenario results (Supplementary Data 2).

### Maintenance cost
We evaluated the potential impact of including a separate maintenance cost in addition to the price-per-scan model. The results suggest that a maintenance cost of £17,000 per year would increase the average cost of using Mia® (from £3,721.00 to £3,725.50). In such a scenario SP would be cost-effective, however under uncertainty (with only a 50% probability). This scenario suggests that the manufacturer may be able to explore different combinations between price-per-scan and maintenance cost and still maintain cost-effectiveness (Supplementary Data 3).

### Table 1 | Base case cost-effectiveness results

| Cost-effectiveness | Costs | QALYs | Incremental Costs | Incremental QALYs | ICER | JackKnife 95% CI | | Net monetary benefit (NMB) | Incremental NMB | Pr of cost-effectiveness |
|---|---|---|---|---|---|---|---|---|---|---|
| Mia® | £3,721 | 16.42258 | | | | | | £324,729.72 | £2.70 | 0.54 |
| Standard practice | £3,726 | 16.42269 | £4.89 | 0.00011 | £44,667 | £5,501 | £73,314 | £324,727.02 | £0.00 | 0.46 |

Estimates over 2000 iterations. Source data are provided as a Source Data file.

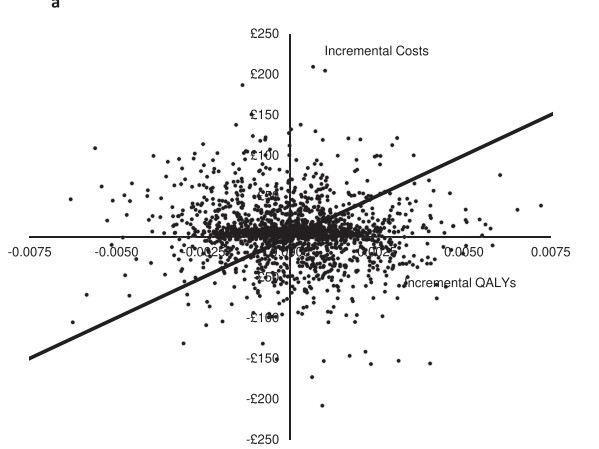

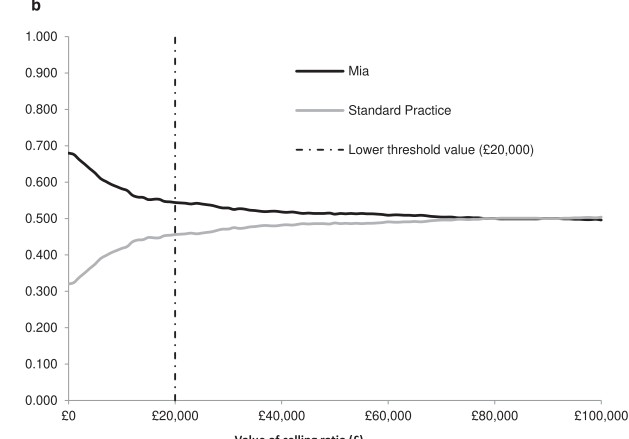

**Fig. 2 | Scatter plot and cost-effectiveness acceptability curve of Mia® versus standard practice. a** Scatter plot Mia® vs standard practice. Iterations represented by the dots that fall below the threshold line indicate iterations where Mia® is cost-effective. Iterations above the threshold line indicate that Standard practice is cost-effective. **b** Cost-effectiveness acceptability curve. The line crossing the £20,000 dot line indicates the probability of cost-effectiveness at that specific threshold. Both the scatter plot and cost-effectiveness acceptability curve are linked as ~54% of the iterations represented by dots shown in the scatter plot are below the threshold line. Source data are provided as a Source Data file.

**Table 2 | Maximum price estimation when Mia®'s sensitivity and specificity are 4.5% and 3.5% lower than SP**

| Intervention | Price per scan | Cost | QALYs | Incremental Costs* | Incremental QALYs* | NMB | INMB | Result |
|---|---|---|---|---|---|---|---|---|
| Mia® Spec 92% Sens 88% | 4.72 | £3,736.82 | 16.4199 | -£2.54 | -0.0015 | £324,661 | -£26.78 | Mia® not cost-effective |
| Mia® Spec 92% Sens 88% | 0.00 | £3,718.08 | 16.4199 | -£21.28 | -0.0015 | £324,680 | -£8.04 | Mia® not cost-effective |
| Standard practice Spec 94% Sens 91% | - | £3,739.36 | 16.4214 | - | - | £324,687 | - | - |

*Mia® interventions are compared against the standard practice (Mia® @4.72 & @£0.00 minus Standard practice). Source data are provided as a Source Data file.

### Impact of Locum (outsourcing of the services)

In this analysis, we explored the impact of the use of a locum to outsource a proportion of the service to meet demand. We ran four different scenarios assuming different locum levels and the price per study charged: a 10% and a 20% of the service covered at £10.30 per reader (£20.60 per complete study); and a 10% of the service covered at ±20% over the price per reader (from £10.30 to £8.24 and £12.36). All scenarios indicate that Mia® is a cost-saving strategy with a probability of over 55% of cost-effectiveness. The latter allows for an increase in the MRP: a 10% increase in the price of the Locum per scan results in an increase of the MRP of 3%; a 10% increase in the use of locum, increases the MRP by 14% (Supplementary Data 4).

## Discussion

The current triannual programme was shown to be only moderately likely cost-effective for use in the NHS by Pharoah et al.[4]. The later results should be taken with caution however as some of the assumptions made by this study (such as no variation in utility with disease stage and no difference in disutility between overdiagnosed cases and actual cases) are likely to have a negative impact on the cost-effectiveness of the breast cancer screening programme. A more recent study set in the Netherlands however indicates that a triannual breast cancer screening was a cost-effective strategy[5].

Our study, however, is aimed at evaluating the use of AI technology as a substitute for one human reader in the NHS national breast screening programme. In our analysis, the base case scenario indicates that Mia® has the potential of being a cost-effective strategy versus current practice and is a feasible technology for use in the NHS. The results are uncertain mainly due to the lack of data to precisely estimate many of the parameters in a complex screening model (such as cancer growth, sensitivity and specificity of both SP and Mia®); however, they suggest that Mia® is at least equivalent to SP. This conclusion is reassuring, mainly as it indicates that such a technology can be used to create longer-term system efficiencies and support the already fragile screening service in the UK (benefits not captured within this analysis). The technology could help the service move from a labour-intensive and time-consuming double-human reader model to a single-human reader and AI model.

The analysis also explored the impact of other variables such as sensitivity, specificity, price-per-scan, set-up, the maintenance cost of the AI technology, and the current service relying on a Locum to meet demand. The results of these analyses suggest that an AI technology such as Mia® needs to be at least equivalent in terms of sensitivity and specificity for it to be a viable alternative; a slightly increased performance on the specificity of Mia®, the proportion of Locum, and its costs have a relevant impact on the cost-effectiveness of the technology allowing for an increase on the MRP; that set-up costs have a small impact on cost-effectiveness and a combination between price-per scan lower than the base case MRP of £5.50 and £17,000 maintenance cost per-year may be possible while maintaining cost-effectiveness (or indifference for Mia®). Given the current financial pressures on the NHS, however, a high set-up cost or yearly maintenance could be a barrier to implementation.

To our knowledge, this is the first study estimating the cost-effectiveness of using AI imaging technology to replace a second human reader in a double-reader national breast cancer screening programme. Other studies have focused on the efficacy of AI technology as equivalent to standard practice in the UK. However, this previous study did not perform cost-effectiveness for the use of AI technology within the NHS context. Furthermore, the system-wide implications, such as the capacity of AI to reduce workload have been explored recently. A systematic review highlighted the ability of AI to reduce workload in three studies, aid diagnosis in six studies, independently mark, and classify suspicious finds comparable with

**Table 3 | Maximum reimbursable price when Mia®'s specificity is at 99% or sensitivity at 79.7%**

| Intervention | Price per scan | Cost | QALYs | Incremental Costs* | Incremental QALYs* | NMB | INMB | Result** |
|---|---|---|---|---|---|---|---|---|
| Mia® Spec 99.0% Sens 75.1% | 4.72 | £3,718.81 | 16.4230 | −£7.90 | 0.00034 | £324,741 | £14.68 | Mia® dominates |
| Mia® Spec 97.3% Sens 79.8% | 4.72 | £3,722.20 | 16.4225 | −£4.51 | −0.00014 | £324,728 | £1.73 | Mia® is cost-effective but small difference indicates likely indifferent |
| Standard practice Spec 97.0% Sens 76.1% | - | £3,726.71 | 16.4227 | | | £324,727 | | |

*Mia® interventions are compared against standard practice.
**Dominates indicates that Mia® is cheaper and more effective.
Source data are provided as a Source Data file.

radiologists in five studies, and potentially be used in a predictive capacity for breast cancer across another seven studies[8].

The model only follows one cohort of women from the age of 19 until death. The actual service, however, will evaluate yearly cohorts of women. This assumption may have an impact on the long-term costs of operating Mia® as two of its main costs components are irrespective of the number of patients scanned. On average the Leeds, UK service offers scans to between 100,000 and 150,000 women per year across all age cohorts and so by running the model for 100,000 women, we aim to minimise this potential impact.

The model was designed and constructed based on the current NHS breast cancer screening pathway. This pathway and the resources required to run the services were constructed with close guidance from the lead specialist in charge of running the NHS Leeds breast cancer screening service. As such the model represents the actual functioning of a generic breast screening service within the UK. However, the unobservable nature of certain key parameters (tumour growth, cancer genesis, breast density) and the lack of large studies estimating the sensitivity and specificity of the service and of the Mia® software hinder the model's ability to match empirical results on the UK's number of cancers detected. As such our model should not be used to estimate these figures. We believe, however, that the latter does not have an impact or undermine the results obtained and presented here as both interventions are evaluated using the sensitivity and specificity as estimated by a one-to-one comparison between standard practice and Mia®, while we used published data on tumour growth, cancer genesis and breast density[3,11,13].

Sensitivity and specificity assumptions for both the Mia® arm and SP are based on demonstrated sensitivity and specificity performance. This information was obtained from a retrospective large-scale evaluation that includes the UK[10]. These parameters are key for estimating the cost-effectiveness of Mia® or any other similar AI technology. Small variations, mainly on the comparative specificity of Mia® against SP could either increase Mia®'s cost-effectiveness or render it as not a viable alternative. While the study that undertook this analysis was considered robust, a larger population in different settings around the UK is required to reduce the uncertainty around these two parameters.

While the stage of the detected cancer is based on the relative size of the tumour, once detected the model only assumes treatment based on the stage upon detection. The model does not follow individual patient's cancer treatment pathway. Similarly, the proportion of non-invasive tumours was based on data published by Tan et al., (2013)[20]. We assume that these tumours do not progress to invasive disease. Although if undetected the tumour will not be treated and will not affect the overall life expectancy of the women, if detected, it was assumed that the patient will undergo treatment and incur costs and quality of life decrements until healed. Compared to Gray et al.[3], our model assumes the same sensitivity for all tumour sizes. Our model may miss some bigger size tumours compared to Grey et al.[3] as this assumption applies to both SP and Mia®'s arm, we

believe that this assumption will not have an impact on the results obtained.

The results are based on all scans being read by a consultant radiologist. Whilst the model can estimate different ratios between radiologists and radiographers, this was not explored due to running time restrictions, but it is expected that a lower ratio between Radiographers/radiologists will increase the standard of practice cost-effectiveness. Conversely, if the service is understaffed and scans are read by more expensive subcontractors this will increase the cost of SP and result in Mia® being the cost-effective strategy.

The model does not account for QALY losses due to a false positive result. Given the small, estimated difference in QALYs between the intervention and control, adding a utility decrement due to a false positive result may lead to an increase in QALYs for Mia® if the technology offers a higher specificity compared to standard practice which may then result in Mia® being the cost-effective strategy even if sensitivity is lower. Similarly, the model does not account for any health benefit due to an increase in a radiologist's time if freed by the Mia® technology which will allow the radiologist to perform other activities. If an additional health benefit is considered, the latter will likely increase the cost-effectiveness of Mia®.

A technology such as Mia® may be a cost-effective alternative for the NHS breast cancer screening programme when performing at similar levels to two independent human readers, providing a set of price points and set-up costs that allowed the programme to be equally costly or cheaper than standard practice. The latter indicates that such a technology has the potential to substitute a human reader to aid services struggling to recruit or meet demand.

## Methods

The DES methodology is an individual agent-based strategy that allowed us to follow individual patients from 19 and through their lifetimes while allowing their individual sampled characteristics (age, cancer genesis, breast density) to determine their future pathway progression.

The developed model was based on Gray et al. (2017)[3] with some simplifications, for instance, our developed model assumes equal sensitivity to all tumour sizes. Other parameters, such as lifetime risk of cancer, cancer genesis, breast density, and survival, were sampled randomly into the population using published data and estimations developed by Gray et al.[3] Once sampled, women follow the pathway described in Fig. 1, and depending on a set of probabilities the patient may develop a tumour that may progress over time. The tumour may be detected in one of the patients' 3-yearly visits depending on the sensitivity and specificity of the readers (adjusted to account for the woman's breast density). If not detected they will continue to grow (in size and stage) based on a cancer growth model from Weedon-Fekjær et al.[11], allowing for the model to simulate the impact of missed diagnosis or false negative results. The latter affects the patient's quality of life and the costs incurred. A detailed description of the process that women follow through the model pathway is described below. The

**Table 4 | Parameters of the model**

| Parameter | Mean/proportion | Standard Error | Distribution | Source/assumptions |
|---|---|---|---|---|
| Starting age | 19 | - | Fixed | Assumed |
| Age of first and last scan | 50–70 | - | Fixed | Based on UK breast screening programme |
| Cancer incidence | Different by age | - | Fixed | Gray et al.[3] based on National Statistic 2012. See supplementary data for values |
| All-cause mortality | Different by age | - | Fixed | Office for National Statistics, (2019). See supplementary data for values |
| **Symptomatic women interim cancer detection by other services (not via the screening service)** | | | | |
| NPI I | 53% | - | Dirichlet non-screening vs screening | Women who seek medical attention due to symptoms between National Cancer Registration & Analysis Service, (2016)[30] |
| NPI II | 80% | - | Dirichlet non-screening vs screening | Women who seek medical attention due to symptoms between National Cancer Registration & Analysis Service, (2016)[30] |
| NPI III | 87% | - | Dirichlet non-screening vs screening | Women who seek medical attention due to symptoms between National Cancer Registration & Analysis Service, (2016)[30] |
| **Breast cancer growth parameters** | | | | |
| Vmax (maximum volume) | 128 mm | - | Fixed | Weedon-Fekjær et al.[11] |
| Vcell (starting size) | 0.25 mm | - | Fixed | Weedon-Fekjær et al.[11] |
| Grow rate | 1.07 | 0.00208 | lognormal | Weedon-Fekjær et al.[11] |
| **Breast density (proportion of women in each group)** | | | | |
| VDG1 | 22% | - | Dirichlet | Wanders et al.[13] |
| VDG2 | 41% | - | Dirichlet | Wanders et al.[13] |
| VDG3 | 29% | - | Dirichlet | Wanders et al.[13] |
| VDG4 | 8% | - | Dirichlet | Wanders et al.[13] |
| **Sensitivity adjustment based on breast density (multiplier)** | | | | |
| VDG1 | 1 | - | Fixed | Adjusted from Gray et al.[3] |
| VDG2 | 0.91 | - | Fixed | Adjusted from Gray et al.[3] |
| VDG3 | 0.81 | - | Fixed | Adjusted from Gray et al.[3] |
| VDG4 | 0.69 | - | Fixed | Adjusted from Gray et al.[3] |
| **Cancer stage** | | | | |
| Non-invasive cancers or DSCI | 0.09 | 0.03 | Beta | Based on Tan et al.[20] as 1 minus the proportion of aggressive cancers |
| NPI I | Size related | - | Dirichlet | Kollias et al.[12] See supplementary data for values |
| NPI II | Size related | - | Dirichlet | Kollias et al.[12] See supplementary data for values |
| NPI III | Size related | - | Dirichlet | Kollias et al.[12] See supplementary data for values |
| Advanced | Sizes related | - | Beta | Gray et al.[3] from the National Health Service audit of screen-detected breast cancers (2013). See supplementary data for values |
| **Cancer survival** | | | | |
| DSCI | As all-cause mortality | - | Fixed | Office for National Statistics, (2019) See supplementary data for values |
| NPI I | 5.413 | - | Exponential | Fong et al.[18]; Gray et al.[3] |
| NPI II | 4.023 | - | Exponential | Fong et al.[18]; Gray et al.[3] |
| NPI III | 2.465 | - | Exponential | Fong et al.[18]; Gray et al.[3] |
| Advanced <50 y | 0.527 | - | Exponential | Fong et al.[18]; Gray et al.[3] |
| Advanced 50–70 y | 0.537 | - | Exponential | Fong et al.[18]; Gray et al.[3] |
| Advanced >70 y | 0.849 | - | Exponential | Fong et al.[18]; Gray et al.[3] |

parameters used to sample for these characteristics can be found in Table 4.

**Movements from the false positive and true negative stages**
If a false positive, the recall stage will reveal that the woman's preliminary diagnosis was mistaken. Therefore, as with the true negative cases, the woman will be issued a 3-year invitation.

**Movements from cancer-detected stage**
As a simplification, the model did not follow the cancer disease progression and assumed that women who have screen-detected cancer will either move to the death from cancer or the death from natural causes end stages. This assumption implies that the women

either die from the disease progression despite treatment or die from natural causes if recovered from cancer or die from natural causes before their cancer progression. Costs and outcomes derived from cancer are considered by stage and severity (more details below).

**Movements from the false negative stage**
As movements from the screen-detected cancer stage, a woman whose cancer was missed could move to death from cancer (if undetected cancer grows symptomatic or progresses to cause death with or without treatment), death from natural causes (if the patient survives cancer or dies before disease progression), and to the invitation to next scan if the undetected cancer growth does not

manifest in a symptomatic disease or if its progression is slow at the time of being the missed diagnosis. Similarly, costs and outcomes derived from cancer are considered by stage and severity (more details below).

### Invitation to next scan

Women are invited to a 3-year scan if their mammography assessment was either, false positive, true negative, or false negative. As the next scan invitation is to occur 3 years after the current scan, the woman may: die from natural causes before the next scan, develop cancer cells (carcinogenesis), or if they already have an undetected cancer growth, this would continue to grow. If alive, after 3 years have elapsed, the woman will receive an invitation to attend the next cycle of mammography according to the NHS scan service. She may accept or reject to attend. If she decides to attend, the process described above will be repeated. If however, the woman misses the 3 years invitation, she will be re-invited after 3 years. As before, during these 3 years, she could either, die from natural causes, develop cancer cells (carcinogenesis) or if already she has an undetected cancer this will continue to grow. For women whose cancer has continued to grow, if it becomes symptomatic, they will have the option to seek medical attention or wait for their next scan appointment.

The described process will continue until the woman dies either from natural causes or due to causes related to a cancer progression.

### Starting age and carcinogenesis

Although the model focuses on NHS breast cancer screening, in which women between the age of 50–70 receive an invitation for a mammography scan every 3 years, the model's starting age is 19. The latter as the model uses the annual probability of a woman developing cancer cells from inception (carcinogenesis). These cells may grow silently for many years before being detected or becoming symptomatic. In some cases, by the time the woman has her first scan, these cells may have increased their size or mutated and develop into an advanced cancer stage. This also allows for the possibility of women developing breast cancer and therefore requiring treatment before their involvement in the national screening programme. The annual probability of developing cancer over the lifetime of the woman was estimated using data from the Office of National Statistics data 2012[21] and based on Gray et al.[3] lifetime risk of developing cancer[3] (Supplementary Data 5).

Despite the model starting age of 19, the model would not account for any costs or QALYs until the women are eligible for the cancer screening programme (age 50). Similarly, only women who have carcinogenesis before the age of 50 but have not sought medical attention before their eligible age for screening are being considered. All-cause mortality before the age of 50 was also not considered.

### Breast cancer growth

Breast cancer growth was estimated based on Gray et al.[3] using the formulations used in the continuous growth model described in Weedon-Fekjær et al.[3,11]. As DES is a discrete-time event model (rather than continuous), we evaluate the cancer growth every 3 years. This information is later used to determine the type and stage of the cancer (Cancer type and stage subsection).

### Detected and undetected tumours

Tumours can either be detected via the breast screening programme, which is triennial, the symptomatic breast service due to breast symptoms, or as an incidental finding by another clinical team. A tumour can also go undetected, by being missed in screening services and being asymptomatic. If the tumour is missed, it either continues to grow until the next screening invitation or in some cases, it will progress further to an advanced condition which could lead to death. These instances will be determined by the stage and severity of the tumour when missed by the screening service. However, if the tumour presence becomes symptomatic the woman may seek medical attention between screening invitations. This was assumed mid-way through the 3-year cycle.

### Cancer type and stage

After a screen-detected tumour, the model evaluates if the detected tumour is a non-invasive, defined as ductal carcinoma in situ (DCIS) or invasive tumour. This is based on the probability of a detected tumour being a DCIS reported by Tan et al.[20]. If non-invasive, the tumour was assumed to remain constant, and although the patient was assumed to receive treatment, it was assumed not to be life-threatening. We assume that these tumours do not progress to invasive disease. Although if undetected the tumour will not be treated and will not affect the overall life expectancy of the women, if detected, it was assumed that the patient will undergo treatment and incur costs and quality of life decrements until healed.

If, however, the tumour is invasive, the model uses its size to determine the stage at diagnosis. We use the Nottingham prognostic index (NPI). This index takes into account the size of the tumour, the number of lymph nodes involved, and the tumour grade. We then use the probability of NPI group membership conditional on tumour size as reported by Kollias et al.[12] (Supplementary Data 6). All invasive cancers irrespectively of the NPI category have a probability of being detected at an advanced stage (or stage IV per the Metastasis classification system[22]). As in Gray et al.[3] these probabilities were also associated with the tumour size based on data from the NHS audit of screen-detected breast cancers (2013)[3].

### Cancer survival and background mortality

The model does not follow cancer progression, it assumes that cancer survivability is linked to the stage of the cancer when detected (cancer detected stage as described above and shown in Fig. 1). Given that the model is evaluating the impact on scan detection rate and accuracy, this assumption implies that cancers detected at earlier stages will have a better prognosis, whilst progression and treatment once detected were assumed to be the same regardless of how or where the cancer was detected. Years remaining of life given the stage at diagnosis were estimated based on Gray et al.[3] using Fong et al.[18] estimates[3,18]. The only exception was DCIS, as these types of cancers were assumed to have the same survival as that of the general population.

General population survivability was based on the Office for National Statistics[23] (Supplementary data 7). Life expectancy for each woman in the cohort is estimated and assigned at the start of the simulation. This is adjusted in the event of the women developing a carcinoma at any stage (NPI I, II or III, or advanced). Therefore, the risk of death from natural causes was present throughout the simulation.

### Sensitivity and specificity

The scan accuracy was measured by the sensitivity and specificity of the independent readers, including those from the Mia® AI technology and arbitration (assumed as an independent Radiologist reader). Data to populate the model was obtained from a retrospective multi-centre, multi-national clinical investigation that evaluated the effectiveness of the Mia® AI technology (Table 5)[10].

Sensitivity was adjusted for breast density. We used the Volpara Density Group (VDG) to determine a woman's breast density. Scans from breasts with a VDG grade 1 will be easier to read than VDG4, hence the sensitivity will be reduced. Since the data we used to populate the model is likely to randomly include women with different breast densities, we assumed no reduction in sensitivity for breast density VG1 with slight incremental penalties for VG2 to VG4. The reductions in sensitivity for breasts with VG2 to VG4 were based on estimates by Gray et al., (2017)[3] (Table 4). To assign women with a breast density we

**Table 5 | Sensitivity and specificity**

| Sensitivity* | Value | Source/assumptions |
|---|---|---|
| **Mia® Arm** | | |
| Reader 1 | 0.761 | One-year sample; Sharma N et al.[10] |
| Mia® as reader 2 | 0.751 | One-year sample; Sharma N et al.[10] |
| **Standard care** | | |
| Reader 1 | 0.760 | One-year sample; Sharma N et al.[10] |
| Reader 2 | 0.760 | One-year sample; Sharma N et al.[10] |
| **Arbitration** | | |
| Mia® Arm | 1.000 | One-year sample; Sharma N et al.[10] |
| Standard care | 1.000 | One-year sample; Sharma N et al.[10] |
| Recall (Biopsy) | 1.000 | Assumed |
| **Specificity*** | | |
| **Mia® Arm** | | |
| Reader 1 | 0.970 | One-year sample; Sharma N et al.[10] |
| Mia® as reader 2 | 0.973 | One-year sample; Sharma N et al.[10] |
| **Standard care** | | |
| Reader 1 | 0.970 | One-year sample; Sharma N et al.[10] |
| Reader 2 | 0.970 | One-year sample; Sharma N et al.[10] |
| **Arbitration** | | |
| Mia® Arm | 0.950 | One-year sample; Sharma N et al.[10] |
| Standard care | 0.950 | One-year sample; Sharma N et al.[10] |
| Recall (Biopsy) | 1.000 | Assumed |

*Sensitivity and specificity were assumed fixed during the Probabilistic sensitivity analysis.

used estimates from Wanders JOP et al. (2017)[13]. This was done randomly at the start of the simulation.

## Costs and quality of Life

Costs and QALYs were estimated in the model depending on each woman's health status. Women who did not have a tumour during their lifetime (either invasive or non-invasive), were assumed to have a quality of life according to their age from the UK tariff[24]. For women with a non-invasive and invasive tumour, we assumed a utility value based on the stage at detection and time from diagnosis: from diagnosis to 6 months, from 6 months to year one, from year one, and up to year 9. If the women survive cancer, their utility will be returned to the national UK average according to their age. This information was based on estimates by Hall et al.[15]. If, however, the woman dies from cancer, during her last year of life, we assumed a utility based on advanced cancer, while the last 6 months will assume utility values for palliative care using estimates from Rautalin et al.[17]. Table 6 contains the utility values used in the model.

Costs of treatment were calculated based on estimates based on grade from Hall et al.[15] and Laudicella et al.[25] These include costs for the first year of treatment: between years 1 to 9, if the woman survives, or to treat advanced cancer and palliative care for their last year of life (Table 7)[15,25]. Although Hall et al.[15] and Laudicella et al.[25] estimates were for grades (I, II, III) we assumed equivalence for NPI I to grade 1, NPI II to grade 2 and NPI III and advanced to grade 3. Costs of the Mia® technology were obtained from discussion with Kheiron Medical Technologies based on their potential price and cost estimates at the time of this study. The implementation of the AI technology requires set-up costs, annual maintenance, and a cost per scan read. All these items were included in the model accordingly. Mia® was assumed to act as a second reader. The cost of the first reader and those of the first and second reader for the standard practices were costed based on the time a consultant radiologist takes to evaluate a scan. These were assumed the same irrespectively of the treatment arm (Mia® or standard practice). The hourly costs of these were obtained from the

PSSRU 2019[19]. The model base case scenario assumes that all scans are read by a consultant radiologist. Other costs included were the costs of the mammography, cost of arbitration, and recall (Table 7). All costs have been updated to 2020 GBP prices using the ONS consumer price index (CPI) health index[16].

## Reporting and sensitivity analysis

We estimate cost-effectiveness using the incremental cost-effectiveness ratio. The model accounts for uncertainty via a probabilistic sensitivity analysis (PSA). The distributions used for this analysis can be found alongside the parameter values described previously and are shown in Table 4, Table 6 and Table 7, respectively. Sensitivity and specificity were assumed fixed, however, to account for the potential influence of these variables, we perform several scenario sensitivity analyses. The main objective of this analysis, however, was to estimate the potential combination of factors that will indicate if the Mia® AI or SP may be cost-effective, depending on the results from the base case analysis.

## First order uncertainty

The DES approach also requires additional considerations to avoid potential bias due to its stochastic nature. Therefore, one model run assumes 100,000 patients. This number was defined along with the mammography service as they indicated to be an average number of patients being seen by the service each year. This large number of patients also has the advantage of limiting the bias of the stochasticity of the DES approach. However, this large number of patients, resulted in long running times, resulting in limiting the number of PSA iterations to 2000. To evaluate the impact of this low number of runs, we estimated JackKnife confidence intervals on the incremental cost-effectiveness ratio to determine if the number of runs was sufficient to determine cost-effectiveness[26]. Additionally, to limit the bias on the deterministic sensitivity analysis, this analysis was performed as probabilistic running 100,000 patients over 2000 iterations. This will avoid potential bias due to a deterministic run.

## Locum assumptions and detailed results

The cost of Locum was assumed based on data published in the Clinical Radiology UK workforce census 2020[27] assuming a £88 million total locum costs paid by the NHS. Around 20% of that sum was expected to be related to the breast cancer service (£17.6 million) to complete an average of 2 million studies (£8.80 per study or £4.40 per reader). This extra cost per reader was added to the current cost per scan incurred by the NHS (£5.90) to give the £10.30 per scan used.

The results of the use of 10% of Locum at the three different prices analysed (£8.80; £10.30 and £12.36) indicate that the use of Mia is cost-effective when compared to standard practice. These results indicate a probability of cost-effectiveness of 56, 58, and 59%, respectively.

Although the cost savings by using Mia® are relatively small (£6.75; £8.38 and £10.02 per woman), the cost difference would allow for an increase in the MRP that Mia® could reach (maximum price at which Mia® would be considered equally cost-effective as standard practice). If 10% of the service relies on Locum for £10.30 per scan the MRP of Mia® would be up to £6.28; to £5.87 for a cost of £8.80 per scan and to £6.69 when the price is set at £12.36 per scan (against the MRP of £5.50 estimated in the base case scenario). The latter can be roughly indicative that an increase of 10% in the cost of the Locum increases the MRP by 3%.

The additional scenario where we assume that the Locum requirements were 20%, indicates Mia® is the cost-effective strategy with a probability of cost-effectiveness of 61%, while the MRP that Mia® could reach to £7.16 per scan. The latter indicates that for every 10 percentual points increase in the use of locum, the MRP will increase by 14%, while the probability of cost-effectiveness of Mia® will increase by three percentage points on average.

## Table 6 | Utility values

| Variable | Mean | Standard Error | Distribution | Source/assumptions |
|---|---|---|---|---|
| **Utility per age group (Women)** | | | | |
| 45–54 | 0.846 | 0.007 | Beta | Szende, Janssen and Cabases, (2014) |
| 55–64 | 0.804 | 0.008 | Beta | Szende, Janssen and Cabases, (2014) |
| 65–74 | 0.76 | 0.009 | Beta | Szende, Janssen and Cabases, (2014) |
| 75+ | 0.692 | 0.01 | Beta | Szende, Janssen and Cabases, (2014) |
| **Utility per cancer stage** | | | | |
| NPI I At 12 months | 0.704 | 0.034 | Beta | From Hall et al.[15]; 6 months estimates assumed up to up to12 months |
| NPI I From 12 months onwards | 0.779 | 0.034 | Beta | From Hall et al.[15]; 15 months assumed from 12 months until discharged or in advanced/palliative care |
| NPI II At 12 months | 0.775 | 0.022 | Beta | From Hall et al.[15]; 6 months estimates assumed up to up to12 months |
| NPI II From 12 months onwards | 0.794 | 0.023 | Beta | From Hall et al.[15]; 15 months assumed from 12 months until discharged or in advanced/palliative care |
| NPI III At 12 months | 0.727 | 0.027 | Beta | From Hall et al.[15]; 6 months estimates assumed up to up to12 months |
| NPI III From 12 months onwards | 0.759 | 0.035 | Beta | From Hall et al.[15]; 15 months assumed from 12 months until discharged or in advanced/palliative care |
| Advanced cancer | 0.74 | 0.26 | Beta | Rautalin et al.[17] |
| Palliative care | 0.51 | 0.29 | Beta | Rautalin et al.[17] |

## Table 7 | Cost values

| Costs | Value | Confidence interval | Distribution | Source/assumptions |
|---|---|---|---|---|
| **Mia® Costs** | | | | |
| Set-up costs | £35,000 | - | Fixed | Kheiron technologies, per Site |
| Maintenance costs | 0 | - | Fixed | Kheiron technologies. Cost is included in the set-up costs |
| Cost per scan | £4.72 | - | Fixed | Kheiron technologies |
| Costs of standard practice | | | | |
| Radiologist | £110 | - | Fixed | Cost per hour of patient contact (PSSRU 2019). Cost per scan: (£5.88) based on 27 scans per working hour |
| Mammography (both breasts) | £184 | - | fixed | 4 scans (one of each breast for two readers) Patient Level Information Costing System (PLICS) Leeds (2020) |
| **Assessments costs (during recall; both interventions)** | | | | |
| Biopsy and exploration | £452 | - | Fixed | Weighted average between different biopsy procedures (core needle biopsy, ultrasound-guided biopsy, fine needle aspiration, etc). National Schedule of NHS Costs Year: 2018-19 - All NHS trusts and NHS foundation trusts - HRG Data |
| Mammography | £46 | - | Fixed | Cost per scan PLICS Leeds (2020) |
| Ultrasound (with no contrast) | £52 | - | Fixed | National Schedule of NHS Costs (2018-19) |
| Triple assessment (overall) | £314 | - | Fixed | Cost-per-patients assumed that the same proportion of patients (14%) will receive either the three assessments (biopsy, mammography, and ultrasound), a combination of two (i.e. biopsy and mammography; biopsy and ultrasound), or just one procedure. |
| Consultant oncologist | £52.9 | - | Fixed | Once cancer is detected. Assumed 20 min consultation (PSSRU 2019). |
| MRI | £148.40 | - | Fixed | To determine the stage of the cancer. National Schedule of NHS Costs–Year 2018-19 |
| **Costs of cancer treatment** | | | | |
| DCIS | £8,968 | £9,692-£10,471 | Lognormal | Assumed as treatment cost for 6 months of a grade 1 cancer. Hall et al.[15] |
| NPI I 1st year | £10,471 | £8,665-£12,615 | Lognormal | Estimated yearly based on Hall et al.[15] |
| NPI II 1st year | £15,484 | £13,548-£17,614 | Lognormal | Estimated yearly based on Hall et al.[15] |
| NPI III 1st year | £21,951 | £19,044-£24,882 | Lognormal | Estimated yearly based on Hall et al.[15] |
| NPI I–II 2nd to 9th year | £2,819 | - | Fixed | Average per year estimated by Laudicella et al.[25] between years 2 and 9 |
| NPI III 2nd to 9th year | £3,815 | - | Fixed | Average per year estimated by Laudicella et al.[25] between years 2 and 9 |
| Advanced cancer costs (6 months) | £14,984 | £13,161-£16,839 | Lognormal | Assumed as 6 months costs of a grade 3 cancer Hall et al.[15] |
| Advanced cancer (12 months) | £21,951 | £19,044-£24,882 | Lognormal | Assumed as yearly costs of grade 3 cancer Hall et al.[15] |
| Palliative care | £14,827 | £1,472-£45,326 | Lognormal | Average cost over 372 survival days. The model uses these figures to estimate the monthly cost of palliative care. |

## Reporting summary

Further information on research design is available in the Nature Portfolio Reporting Summary linked to this article.

## Data availability

The analysis was conducted as described in the manuscript based on an already published model cited in the manuscript. All data used to populate the model are publicly available and referenced. All parameters and their values can be found in Tables 4–7. The data sets used can be found in the Supplementary data. Data generated from the analysis and used to construct Tables 1–3 and Fig. 2 have been deposited in Figshare as an Excel file under the name Source Data Mia Model (https://doi.org/10.6084/m9.figshare.23295194).

## Code availability

The model was constructed using the Simul8® software Educational site license[28]. The model is intended for research purposes and its use is limited to this purpose. The model can be provided via a request to the corresponding author to a.vargas-palacios@leeds.ac.uk, however, access will only be granted if the intended use for the model is limited to academic/reproducibility proposes only. When requesting access to the model please indicate clearly the intended use and target audience. Response to requests will be dealt with within 5 UK working days. If approved, there are no restrictions to the use of the data contained in the model. The visual logic code of the model, however, can be found in Zenodo.org (https://doi.org/10.5281/zenodo.8192843)[29]. This code can be used to replicate the model using Simul8®.

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

## Acknowledgements

The authors would like to thank Kheiron Medical Technologies for their support in the development of this manuscript. This work was supported by a grant awarded by Innovate UK [project number 104806] to N.S. and G.S.S. The authors would also like to state that the contents of this manuscript and the presentation of the results and conclusions are their opinions and not necessarily those of Kheiron Medical Technologies.

## Author contributions

A.V.-P. led the model design, model development, and the writing of the final manuscript. N.S. led the clinical input, offered clinical guidance,

and contributed to the writing of the final manuscript. G.S.S was the project leader and contributed to the model design and writing of the final manuscript.

## Competing interests

The authors declare no competing interests.
