## [Peer Review File · Nature Communications]

Cost-effectiveness requirements for implementing artificial intelligence technology in the Women's UK breast cancer screening serviceREVIEWER COMMENTS

Reviewer #1 (Remarks to the Author):

This study reports a cost-effectiveness analysis using a discrete event simulation comparing current standard breast screening with double-reading mammography to single reading mammography aided by Mia.

The results have face validity, very small difference in sensitivity and specificity translate into very small differences in cost and QALYs. There is a cost-saving with Mia that is largely determined by the difference in costs of double reading vs single reading + Mia (with some additional small saving from slightly greater specificity). This result is uncertain because the impact of marginal changes in sensitivity/specificity of mammography on screening programme outcomes are uncertain. That should be true both for this discrete event simulation model or any other screening model. This is because there is not sufficient data to precisely estimate many of the parameters in a complex screening model. Additionally, there may be structural uncertainty to consider, while model validation is limited due to few large experimental studies of screening.

There is one statement that I believe is incorrect and should be amended:

P7: "The results are uncertain due to the stochastic nature of the DES modelling technique; however, they suggest that Mia is at least equivalent to SP."

If the results are uncertain only due to the stochastic nature of DES then a larger number of individuals should be simulated until this is not the case. However, I think the uncertainty in the PSA is not due to using a DES but due to uncertainty in many of the model parameters. In addition, because sensitivity and specificity are similar for Mia and SP, even with a small amount of uncertainty, the results will flip between cost-effective and not cost-effective in PSA iterations (although the differences in costs and QALYs between alternatives will still be small). The statement should be amended to highlight that uncertainty in the results is due to uncertainty in some of parameters of a complex screening model (e.g. there is limited data to estimate a cancer growth model).

I have some questions about the methodology that may merit minor revision:

The discrete event simulation model as adapted from a previously published model. It is important to understand any modifications to this model and their justification.

In the Gray et al model sensitivity varies by lesion size. How was the single estimate of sensitivity applied in this adapted model? Was this assumed to be equal for all tumour sizes at screening events? This could be clarified in the main text or in the appendix text of the detailed model description.

Simulating a 100K patient cohort – is this a large enough simulated sample to obtain a stable result?

I would like more detail on the cost per scan for Radiologists. Is this per reading? So that SP reading costs are $£5.90 \times 2 = £11.80$ while reading costs with Mia are $£5.90 + £4.72 = £10.32$. This could be clarified on page 4. The radiologist's unit cost and time per reading could also be stated here.

The introduction and discussion generally present appropriate conclusions and note the key limitations of this type of modelling. There are a couple of points that could be considered further depending on what the author's consider the most important messages.

The abstract and introduction mentions the particular pressure on radiologist's time. This may be something that goes over and above the direct costs to the NHS of employing the radiologists. Because of the constraints in the supply of radiologists (like many other medical specialists) the full loss of value to the health service from radiologists time may not accounted for in the costs of employment alone. This means a limitation is that there may be an additional health benefit, related to increased radiologist time, that is not valued in this model.

While there is a great deal of uncertainty in the results both for costs and health benefits there is one aspect of the analysis that is much more certain. This is the budget impact for the screening programme. This does not depend on understanding the downstream impacts of small changes in sensitivity and specificity and therefore not subject to uncertainty arising from the complex screening model.

Reviewer #2 (Remarks to the Author):

1. This paper reports on a cost-effectiveness modelling exercise on use of an artificial intelligence tool within the NHS Breast Screening Programme in the UK. The conclusion is that the use of the AI tool is likely to be cost-effective. While this is probably true, I did have a concern about calibration of the model. Does the model predict the relative effect of current practice on breast cancer deaths as found in the the trials- around a 20% reduction with the offer of screening and somewhat more than a 30%

reduction with actually being screened, amounting to around 6 lives saved per thousand women screened in the UK programme?

2. Similarly do the assumed costs tally with the cost of the programme as determined by the National Audit Office?

3. Reference 2 is problematic. It assumes no variation in utility with disease stage and not difference in disutility between overdiagnosed cases and 'real' cases. It also confuses screening with invitation. I suggest reference to the Erasmus evaluation of cost-effectiveness instead.

4. In figure 1, should "death from cancer" be "death from breast cancer", and should "death from natural causes" be "death from other causes"?

REVIEWER COMMENTS

Reviewer #1 (Remarks to the Author):

This study reports a cost-effectiveness analysis using a discrete event simulation comparing current standard breast screening with double-reading mammography to single reading mammography aided by Mia.

The results have face validity, very small difference in sensitivity and specificity translate into very small differences in cost and QALYs. There is a cost-saving with Mia that is largely determined by the difference in costs of double reading vs single reading + Mia (with some additional small saving from slightly greater specificity). This result is uncertain because the impact of marginal changes in sensitivity/specificity of mammography on screening programme outcomes are uncertain. That should be true both for this discrete event simulation model or any other screening model. This is because there is not sufficient data to precisely estimate many of the parameters in a complex screening model. Additionally, there may be structural uncertainty to consider, while model validation is limited due to few large experimental studies of screening.

There is one statement that I believe is incorrect and should be amended:

P7: "The results are uncertain due to the stochastic nature of the DES modelling technique; however, they suggest that Mia is at least equivalent to SP."

If the results are uncertain only due to the stochastic nature of DES then a larger number of individuals should be simulated until this is not the case. However, I think the uncertainty in the PSA is not due to using a DES but due to uncertainty in many of the model parameters. In addition, because sensitivity and specificity are similar for Mia and SP, even with a small amount of uncertainty, the results will flip between cost-effective and not cost-effective in PSA iterations (although the differences in costs and QALYs between alternatives will still be small). The statement should be amended to highlight that uncertainty in the results is due to uncertainty in some of parameters of a complex screening model (e.g. there is limited data to estimate a cancer growth model).

Thank you for your comments. We have changed the statement to: "The results are uncertain due to the uncertainty of some of the model parameters within a complex screening model (such as cancer growth) and the similarity in the sensitivity and specificity of SP and Mia); however, they suggest that Mia is at least equivalent to SP."

I have some questions about the methodology that may merit minor revision:

The discrete event simulation model as adapted from a previously published model. It is important to understand any modifications to this model and their justification.

We have added further details to the methods section to address this concern.

In the Gray et al model sensitivity varies by lesion size. How was the single estimate of sensitivity applied in this adapted model? Was this assumed to be equal for all tumour sizes at screening events? This could be clarified in the main text or in the appendix text of the detailed model description.

Thank you for your comments. Sensitivity was assumed equal for all tumour sizes and we have now added this detail into both the main text and the methods section of the paper.

Simulating a 100K patient cohort – is this a large enough simulated sample to obtain a stable result?

Results are stable when 100,000 patients are considered, the difference between costs and QALYS between interventions from 75,000 patients remains almost identical between them (Mia is slightly less costly and slightly less effective) with no meaningful impact on the overall results presented. Furthermore, changes in costs and QALYs within intervention do not change significantly from 75,000 patients. No changes have been made to the manuscript.

I would like more detail on the cost per scan for Radiologists. Is this per reading? So that SP reading costs are $£5.90 \times 2 = £11.80$ while reading costs with Mia are $£5.90 + £4.72 = £10.32$. This could be clarified on page 4. The radiologist's unit cost and time per reading could also be stated here.

As stated by the reviewer the cost per scan is £5.90 and £4.72 for SP and the Mia arm. The cost per reading (per woman) is £11.80 and £10.32 respectively. This has been clarified in the main text of the manuscript.

The introduction and discussion generally present appropriate conclusions and note the key limitations of this type of modelling. There are a couple of points that could be considered further depending on what the author's consider the most important messages.

The abstract and introduction mentions the particular pressure on radiologist's time. This may be something that goes over and above the direct costs to the NHS of employing the radiologists. Because of the constraints in the supply of radiologists (like many other medical specialists) the full loss of value to the health service from radiologists time may not accounted for in the costs of employment alone. This means a limitation is that there may be an additional health benefit, related to increased radiologist time, that is not valued in this model.

Thank you for your suggestions. We have included this in the discussion section of the main document.

While there is a great deal of uncertainty in the results both for costs and health benefits there is one aspect of the analysis that is much more certain. This is the budget impact for the screening programme. This does not depend on understanding the downstream impacts of small changes in sensitivity and specificity and therefore not subject to uncertainty arising from the complex screening model.

Thanks for your suggestion. Given the small differences in the total costs between SP and Mia, we consider that the budget impact of implementing Mia technology would be minimal, despite the set-up costs of the technology. We have added this to the discussion section of the main text

Reviewer #2 (Remarks to the Author):

1. This paper reports on a cost-effectiveness modelling exercise on use of an artificial intelligence tool within the NHS Breast Screening Programme in the UK. The conclusion is that the use of the AI tool is likely to be cost-effective. While this is probably true, I did have a concern about calibration of the model. Does the model predict the relative effect of current practice on breast cancer deaths as found in the trials- around a 20% reduction with the offer of screening and somewhat more than a 30% reduction with actually being screened, amounting to around 6 lives saved per thousand women screened in the UK programme?

The model is not set up to evaluate the outcomes of screening versus no screening, so we are unable to replicate the above results. The cost-effectiveness of the NHS screening service has been previously evaluated (Pharoah P D P, et al. 2013). The aim of our analysis was to investigate under which conditions an AI technology would be feasible for use in the current NHS system as a replacement for one of the two human readers required to run the service, focusing on cost and QALY outcomes.

Calibration (or external validation) in this case is not possible due to the number of uncertain parameters and limited available data on them and the sensitivity and specificity of both the readers and Mia technology (as highlighted by Reviewer 1). In our model, as both interventions are evaluated using the sensitivity and specificity as estimated by single a one-to-one comparison between the SP and Mia, we believe that this does not undermine the results and recommendations obtained.

2. Similarly do the assumed costs tally with the cost of the programme as determined by the National Audit Office?

Our model does not estimate the cost of the programme alone, as our estimates include those of the treatment of the detected/undetected cancer and end-of-life treatment. The costs used to estimate costs of the personnel and procedures involved in the programme were estimated using costs reported by the NHS using publicly available sources (PSSRU, the National Schedule of NHS costs and the Patient level information and costing system). Those costs were considered per unit of activity and include wages, training, travel and capital and management and staff overheads (as reported by PSSRU), but do not include the cost of the equipment used as these are considered sunk (except set-up costs for Mia). As such we do not expect our costs to tally those determined by the national audit office.

3. Reference 2 is problematic. It assumes no variation in utility with disease stage and not difference in disutility between overdiagnosed cases and 'real' cases. It also confuses screening with invitation. I suggest reference to the Erasmus evaluation of cost-effectiveness instead.

Thanks for your suggestion, we are happy to add this reference to the manuscript. We acknowledge the issues with reference 2 but would like to keep it as it is an evaluation done specifically for the NHS in the UK.

4. In figure 1, should "death from cancer" be "death from breast cancer", and should "death from natural causes" be "death from other causes"?

Thanks for your suggestions, this has been amended

REVIEWER COMMENTS

Reviewer #1 (Remarks to the Author):

Thank you for responding to my comments on the previous version. The responses and revisions have addressed the issues raised. I think the additional information makes clear the intuition of the result. It has clarified the points of the modelling that were not fully explicit and replicable previously.

I have no further comments on the new version of the manuscript.

Reviewer #2 (Remarks to the Author):

1. The authors have pretty much dismissed most of my previous comments, notably on calibration. Changes made seem cosmetic at best. While the specific calibrations I requested may not be practicable with this model, it remains a matter of some importance to check whether the model agrees with empirical reality. I would suggest returning to this issue.
2. While the Pharoah et al paper does address the NHS programme, it is also unreliable for reasons I gave in my previous report. I suggest dropping reference to it or at least saying why one might have reservations about it.

- 1) The authors have pretty much dismissed most of my previous comments, notably on calibration. Changes made seem cosmetic at best. While the specific calibrations I requested may not be practicable with this model, it remains a matter of some importance to check whether the model agrees with empirical reality. I would suggest returning to this issue.

We did not intend for the reviewer to feel that we dismissed their initial comments regarding our manuscript. We would like to thank them for their review and especially thank them for recognising the difficulties of calibrating our developed model to match empirical data. We recognise that this is a limitation of our study and have addressed it as a discussion point in the paper as follows: *The model was designed and constructed based on the current NHS breast cancer screening pathway. This pathway and the resources required to run the services were constructed with close guidance from a lead specialist in charge of running the NHS Leeds breast cancer screening service. As such the model represents the actual functioning of a generic breast screening service within the UK. However, the unobservable nature of certain key parameters (tumour growth, cancer genesis, breast density) and the lack of large studies estimating the sensitivity and specificity of the service and of the Mia software hinder the model's ability to match empirical results on the UK's number of cancers detected. As such our model should not be used to estimate these figures. We believe, however, that the latter does not have an impact or undermine the results obtained and presented here as both interventions are evaluated using the sensitivity and specificity as estimated by a one-to-one comparison between standard practice and Mia, while we used published data on tumour growth, cancer genesis and breast density.*

We would like to discuss further the limitations of calibrating our model. A calibration process such as that required to match empirical data would require a Markov Chain Monte Carlo (MCMC) calibration process. This process is computationally expensive as it requires at least running the model for 50,000 iterations. Given the complexity of the model we have built (with more than 100 variables and following every individual woman from the age of 19 for their lifetime) each iteration requires over 2 minutes to be completed. As such attempting a calibration would require an excessive amount of computing power and resources. The latter combined with the number of variables required to be subject to calibration (at least 5) will considerably reduce the probability of achieving convergence.

- 2) While the Pharoah et al paper does address the NHS programme, it is also unreliable for reasons I gave in my previous report. I suggest dropping reference to it or at least saying why one might have reservations about it.

Thank you for your suggestion. As requested by the reviewer we have added the concerns raised regarding the Pharoah et al paper in our discussion as follows: *The current triannual program was shown only to be only moderately likely cost-effective for use in the NHS by Pharoah et al 2013⁴. The later results should be taken with caution however as some of the assumptions made by this study (such as no variation in utility with disease stage and no difference in disutility between overdiagnosed cases and actual cases) are likely to have a negative impact on the cost-effectiveness of the breast cancer screening programme. A more recent study set in the Netherlands however indicates that a triannual breast cancer screening was a cost-effective strategy.⁵*

We believe that it is better to include it rather than ignoring its existence by removing it despite the potential limitation of this paper. We feel it is important to reference it as it evaluates the cost-

effectiveness of the NHS programme and have included comments raised to place it into context. We hope that the reviewer is happy with our approach but if the editor decides otherwise, we would be willing to drop this reference if both the reviewer and the editor feels that our clarification is not enough or if the use of this reference with the clarification would confer the wrong message to readers of our paper.